# Scalable Graph Neural Networks via Bidirectional Propagation

**Ming Chen**
Renmin University of China
chennnming@ruc.edu.cn

**Zhewei Wei**\*
Renmin University of China
zhewei@ruc.edu.cn

**Bolin Ding**
Alibaba Group
bolin.ding@alibaba-inc.com

**Yaliang Li**
Alibaba Group
yaliang.li@alibaba-inc.com

**Ye Yuan**
Beijing Institute of Technology
yuan-ye@bit.edu.cn

**Xiaoyong Du**
Renmin University of China
duyong@ruc.edu.cn

**Ji-Rong Wen**
Renmin University of China
jrwen@ruc.edu.cn

## Abstract

Graph Neural Networks (GNN) is an emerging field for learning on non-Euclidean data. Recently, there has been increased interest in designing GNN that scales to large graphs. Most existing methods use "graph sampling" or "layer-wise sampling" techniques to reduce training time. However, these methods still suffer from degrading performance and scalability problems when applying to graphs with billions of edges. This paper presents GBP, a scalable GNN that utilizes a localized bidirectional propagation process from both the feature vectors and the training/testing nodes. Theoretical analysis shows that GBP is the first method that achieves sub-linear time complexity for both the precomputation and the training phases. An extensive empirical study demonstrates that GBP achieves state-of-the-art performance with significantly less training/testing time. Most notably, GBP can deliver superior performance on a graph with over 60 million nodes and 1.8 billion edges in less than half an hour on a single machine.

## 1 Introduction

Recently, the field of Graph Neural Networks (GNNs) has drawn increasing attention due to its wide range of applications such as social analysis [23, 20, 28], biology [10, 26], recommendation systems [36], and computer vision [39, 7, 13]. Graph Convolutional Network (GCN) [15] adopts a message-passing approach and gathers information from the neighbors of each node from the previous layer to form the new representation. The vanilla GCN uses a full-batch training process and stores each node's representation in the GPU memory, which leads to limited scalability. On the other hand, training GCN with mini-batches is difficult, as the neighborhood size could grow exponentially with the number of layers.

These techniques can be broadly divided into three categories: 1) Layer-wise sampling methods: GraphSAGE [11] proposes a neighbor-sampling method to sample a fixed number of neighbors for each node. VRGCN [6] leverages historical activations to restrict the number of sampled nodes

and reduce the variance of sampling. FastGCN [5] samples nodes of each layer independently based on each node's degree and keeps a constant sample size in all layers to achieve scales linearly. LADIES [40] further proposes a layer-dependent sampler to constrain neighbor dependencies, which guarantees the connectivity of the sampled adjacency matrix. 2) Graph Sampling methods: Cluster-GCN [8] builds a complete GCN from clusters in each mini-batch. GraphSAINT [37] proposes several light-weight graph samplers and introduces a normalization technique to eliminate biases of mini-batch estimation. 3) Linear Models: SGC [30] computes the product of the feature matrix and the $k$-th power of the normalized adjacency matrix during the preprocessing step and performs standard logistic regression to remove redundant computation. PPRGo [4] uses Personalized PageRank to capture multi-hop neighborhood information and uses a forward push algorithm [2] to accelerate computation.

While the above methods significantly speed up the training time of GNNs, they still suffer from three major drawbacks. First of all, the time complexity is linear to $m$, the number of edges in the graph. In theory, this complexity is undesirable for scalable GNNs. Secondly, as we shall see in Section 4, the existing scalable GNNs, such as GraphSAINT, LADIES, and SGC, fail to achieve satisfying results in the semi-supervised learning tasks. Finally and most importantly, none of the existing methods can offer reliable performance on billion-scale graphs.

**Our contributions.** In this paper, we first carefully analyze the theoretical complexity of existing scalable GNNs and explain why they cannot scale to graphs with billions of edges. Then, we present GBP (**G**raph neural network via **B**idirectional **P**ropagation), a scalable Graph Neural Network with sub-linear time complexity in theory and superior performance in practice. GBP performs propagation from both the feature vector and the training/testing nodes, yielding an unbiased estimator for each representation. Each propagation is executed in a localized fashion, leading to sub-linear time complexity. After the bidirectional propagation, each node's representation is fixed and can be trivially trained with mini-batches. The empirical study demonstrates that GBP consistently improves the performance and scalability across a wide variety of datasets on both semi-supervised and fully-supervised tasks. Finally, we present the first empirical study on a graph with over 1.8 billion edges. The result shows that GBP achieves superior results in less than 2,000 seconds on a moderate machine.

## 2 Theoretical analysis of existing methods

In this section, we review some of the recent scalable GNNs and analyze their theoretical time complexity. We consider an undirected graph $G=(V, E)$, where $V$ and $E$ represent the set of vertices and edges, respectively. For ease of presentation, we assume that $G$ is a *self-looped* graph [15], with a self-loop attached to each node in $V$. Let $n = |V|$ and $m = |E|$ denote the number of vertices and edges in $G$, respectively. Each node is associated with an $F$-dimensional feature vector and we use $\mathbf{X} \in \mathcal{R}^{n \times F}$ to denote the feature matrix. We use $\mathbf{A}$ and $\mathbf{D}$ to represent the adjacency matrix and the diagonal degree matrix of $G$, respectively. For each node $u \in V$, $N(u)$ is the set of neighbors of $u$, and $d(u) = |N(u)|$ is the degree of $u$. We use $d = \frac{m}{n}$ to denote the average degree of $G$. Following [15], we define the normalized adjacency matrix of $G$ as $\tilde{\mathbf{A}} = \mathbf{D}^{-1/2}\mathbf{A}\mathbf{D}^{-1/2}$. The $(\ell + 1)$-th layer $\mathbf{H}^{(\ell+1)}$ of the vanilla GCN is defined as

$$\mathbf{H}^{(\ell+1)} = \sigma\left(\tilde{\mathbf{A}}\mathbf{H}^{(\ell)}\mathbf{W}^{(\ell)}\right), \qquad (1)$$

where $\mathbf{W}^{(\ell)}$ is the learnable weight matrix and $\sigma(\cdot)$ is the activation function. The training and inference time complexity of a GCN with $L$ layers can be bounded by $O\left(LmF + LnF^2\right)$, where $O\left(LmF\right)$ is the total cost of the sparse-dense matrix multiplication $\tilde{\mathbf{A}}\mathbf{H}^{(\ell)}$, and $O(LnF^2)$ is the total cost of the feature transformation by applying $\mathbf{W}^{(\ell)}$. At first glance, $O(LnF^2)$ seems to be the dominating term, as the average degree $d$ on scale-free networks is usually much smaller than the feature dimension $F$ and hence $LnF^2 > LndF = LmF$. However, in reality, feature transformation can be performed with significantly less cost due to better parallelism of dense-dense matrix multiplications. Consequently, $O\left(LmF\right)$ is the dominating complexity term of GCN and performing full neighbor propagation $\tilde{\mathbf{A}}\mathbf{H}^{(\ell)}$ is the main bottleneck for achieving scalability.

In order to speed up GCN training, a few recent methods use various techniques to approximate the full neighbor propagation $\tilde{\mathbf{A}}\mathbf{H}^{(\ell)}$ and enable mini-batch training. We divide these methods into three categories and summarize their time complexity in Table 1.

**Layer-wise sampling** methods sample a subset of the neighbors at each layer to reduce the neighborhood size. GraphSAGE [11] samples $s_n$ neighbors for each node and only aggregates the embeddings from the sampled nodes. With a batch-size $b$, the cost of feature propagation is bounded by $O(bs_n^L F)$, and thus the total per-epoch cost of GraphSAGE is $O(ns_n^L F + ns_n^{L-1}F^2)$. This complexity grows exponentially with the number of layers $L$ and is not scalable on large graphs. Another work [6] based on node-wise sampling further reduces sampling variance to achieve a better convergence rate. However, it suffers from worse time and space complexity. FastGCN [5] and LADIES [40] restrict the same sample size across all layers to limit the exponential expansion. If we use $s_l$ to denote the number of nodes sampled per layer, the per-batch feature propagation time is bounded by $O(Ls_l^2 F)$. Since $\frac{n}{s_l}$ batches are needed in an epoch, it follows that the per-epoch forward propagation time is bounded by $O(Lns_l F + LnF^2)$. Mini-batch training significantly accelerates the training process of the layer-wise sampling method. However, the training time complexity is still linear to $m$ as the number of samples $s_l$ is usually much larger than the average degree $d$. Furthermore, it has been observed in [8] that the overlapping nodes in different batches will lead to high computational redundancy, especially in fully-supervised learning.

**Graph sampling** methods sample a sub-graph at the beginning of each batch and perform forward propagation on the same subgraph across all layers. Cluster-GCN [8] uses graph clustering techniques [14] to partition the original graph into several sub-graphs, and samples one sub-graph to perform feature propagation in each mini-batch. In the worst case, the number of clusters in the graph is 1, and Cluster-GCN essentially becomes vanilla GCN in terms of time complexity. GraphSAINT [37] samples a certain amount of nodes and uses the induced sub-graph to perform feature propagation in each mini-batch. Let $b$ denote the number of sampled node per-batch, and $\frac{n}{b}$ denote the number of batches. Given a sampled node $u$, the probability that a neighbor of $u$ is also sampled is $b/n$. Therefore, the expected number of edges in the sub-graph is bounded by $O(b^2 d/n)$. Summing over $\frac{n}{b}$ batches follows that the per-epoch feature propagation time of GraphSAINT is bounded by $O(LbdF)$, which is sub-linear to the number of edges in the graph. However, GraphSaint requires a full forward propagation in the inference phase, leading to the $O(LmF + LnF^2)$ time complexity.

**Linear model** removes the non-linearity between each layer in the forward propagation, which allows precomputation of the final feature propagation matrix and result in an optimal training time complexity of $O(nF^2)$. SGC [30] repeatedly perform multiplication of normalized adjacency matrix $\tilde{\mathbf{A}}$ and feature matrix $\mathbf{X}$ in the precomputation phase, which requires $O(LmF)$ time. PPRGo [4] calculates approximate the Personalized PageRank (PPR) matrix $\sum_{\ell=0}^{\infty}\alpha(1-\alpha)^\ell\tilde{\mathbf{A}}^\ell$ by forward push algorithm [2] and then applies the PPR matrix to the feature matrix $X$ to derive the propagation matrix. Let $\varepsilon$ denote the error threshold of the forward push algorithm, the precomputation cost is bounded by $O(\frac{m}{\varepsilon})$. A major drawback of PPRGo is that it takes $O(\frac{n}{\varepsilon})$ space to store the PPR matrix, rendering it infeasible on billion-scale graphs.

Table 1: Summary of time complexity for GNN training and inference.

| Method | Precomputation | Training | Inference |
| --- | --- | --- | --- |
| GCN | - | $O\left(LmF + LnF^2\right)$ | $O\left(LmF + LnF^2\right)$ |
| GraphSAGE | - | $O\left(ns_n^L F + ns_n^{L-1}F^2\right)$ | $O\left(ns_n^L F + ns_n^{L-1}F^2\right)$ |
| FastGCN | - | $O\left(Lns_l F + LnF^2\right)$ | $O\left(Lns_l F + LnF^2\right)$ |
| LADIES | - | $O\left(Lns_l F + LnF^2\right)$ | $O\left(Lns_l F + LnF^2\right)$ |
| SGC | $O\left(LmF\right)$ | $O\left(nF^2\right)$ | $O\left(nF^2\right)$ |
| PPRGo | $O\left(\frac{m}{\varepsilon}\right)$ | $O\left(nKF + LnF^2\right)$ | $O\left(nKF + LnF^2\right)$ |
| Cluster-GCN | $O\left(m\right)$ | $O\left(LmF + LnF^2\right)$ | $O\left(LmF + LnF^2\right)$ |
| GraphSAINT | - | $O\left(LbdF + LnF^2\right)$ | $O\left(LmF + LnF^2\right)$ |
| GBP (This paper) | $O\left(LnF + L\frac{\sqrt{m\lg n}}{\varepsilon}F\right)$ | $O\left(LnF^2\right)$ | $O\left(LnF^2\right)$ |

**Other related work**. Another line of research devotes to attention model [29, 27, 22], where the adjacency matrix of each layer is replaced by a learnable attention matrix. GIN [31] studies the expressiveness of GNNs, and shows that GNNs are not better than the Weisfeiler-Lehman test in

distinguishing graph structures. GDC [17] proposes to replace the graph convolutional matrix $\tilde{\mathbf{A}}$ with a graph diffusion matrix, such as the Heat Kernel PageRank or the Personalized PageRank matrix. Mixhop [1] mixes higher-order information to learn a wider class of representations. JKNet [32] explores the relationship between node influence and random walk in GNNs. DropEdge [24] and PairNorm [38] focus on over-smoothing problem and improve performance on GCNs by increasing the number of layers. These works focus on the effectiveness of GNNs; Thus, they are orthogonal to this paper's contributions.

## 3 Bidirectional Propagation Method

**Generalized PageRank.** To achieve high scalability, we borrow the idea of decoupling prediction and propagation from SGC [30] and APPNP [16]. In particular, we precompute the feature propagation with the following *Generalized PageRank* matrix [21]:

$$\mathbf{P} = \sum_{\ell=0}^{L} w_\ell \mathbf{T}^{(\ell)} = \sum_{\ell=0}^{L} w_\ell \left( \mathbf{D}^{r-1} \mathbf{A} \mathbf{D}^{-r} \right)^\ell \cdot \mathbf{X}, \tag{2}$$

where $r \in [0,1]$ is the convolution coefficient, $w_\ell$'s are the weights of different order convolution matrices that satisfy $\sum_{\ell=0}^{\infty} w_\ell \leq 1$, and $\mathbf{T}^{(\ell)} = \left( \mathbf{D}^{r-1} \mathbf{A} \mathbf{D}^{-r} \right)^\ell \cdot \mathbf{X}$ denotes the $\ell$-th step propagation matrix. After the feature propagation matrix $\mathbf{P}$ is derived, we can apply a multi-layer neural network with mini-batch training to make the prediction. For example, for multi-class classification tasks, a two-layer GBP model makes prediction with $Y = SoftMax \left( \sigma \left( \mathbf{P} \mathbf{W_1} \right) \mathbf{W_2} \right)$ where $\mathbf{W_1}$ and $\mathbf{W_2}$ are the learnable weight matrices, and $\sigma$ is the activation function.

We note that equation (2) can be easily generalized to various existing models. By setting $r = 0.5, 0$ and 1, the convolution matrix $\mathbf{D}^{r-1} \mathbf{A} \mathbf{D}^{-r}$ represents the symmetric normalization adjacency matrix $\mathbf{D}^{-1/2} \mathbf{A} \mathbf{D}^{-1/2}$ [15, 30, 16], the transition probability matrix $\mathbf{A} \mathbf{D}^{-1}$ [11, 8, 37], and the reverse transition probability matrix $\mathbf{D}^{-1} \mathbf{A}$ [32], respectively. We can also manipulate the weights $w_\ell$ to simulate various diffusion processes as in [17]. However, we will mainly focus on two setups in this paper: 1) $w_\ell = \alpha(1-\alpha)^\ell$ for some constant decay factor $\alpha \in (0,1)$, in which case $\mathbf{P}$ becomes the Personalized PageRank used in APPNP and PPRGo [16, 17, 4]; 2) $w_\ell = 0$ for $\ell = 0, \ldots, L-1$ and $w_L = 1$, in which case $\mathbf{P}$ degenerates to the $L$-th transition probability matrix in SGC [30].

---

**Algorithm 1:** Bidirectional Propagation Algorithm

---

**Input:** Graph $G$, level $L$, training set $V_t$, weight coefficients $w_\ell$, convolutional coefficient $r$, threshold $r_{max}$, number of walks per node $n_r$, feature matrix $\mathbf{X}_{n \times F}$
**Output:** Embedding matrix $\mathbf{P}_{n \times F}$

1   $\mathbf{S}^{(\ell)} \leftarrow Sparse\left(\mathbf{0}_{n \times n}\right)$ for $\ell = 0, \ldots, L$;
2   **for** *each node $s \in V_t$* **do**
3      Generate $n_r$ random walks from $s$, each of length $L$;
4      **if** *The $j$-th random walk visits node $u$ at the $\ell$-th step, $\ell = 0, \ldots, L$, $j = 1, \ldots, n_r$* **then**
5          $\mathbf{S}^{(\ell)}(s, u)$ += $\frac{1}{n_r}$;

6   $\mathbf{Q}^{(\ell)}, \mathbf{R}^{(\ell)} \leftarrow Sparse\left(\mathbf{0}_{n \times F}\right)$ for $\ell = 1, \ldots, L$;
7   $\mathbf{Q}^{(0)} \leftarrow \mathbf{0}_{n \times F}$ and $\mathbf{R}^{(0)} \leftarrow ColumnNormalized\left(\mathbf{D}^{-r}\mathbf{X}\right)$;
8   **for** $\ell$ *from* $0$ *to* $L-1$ **do**
9      **for** *each $u \in V$ and $k \in \{0, \ldots, F-1\}$ with $\left| \mathbf{R}^{(\ell)}(u,k) \right| > r_{max}$* **do**
10          **for** *each $v \in \mathcal{N}(u)$* **do**
11              $\mathbf{R}^{(\ell+1)}(v,k)$ += $\frac{\mathbf{R}^{(\ell)}(u,k)}{d(v)}$;
12          $\mathbf{Q}^{(\ell)}(u,k) \leftarrow \mathbf{R}^{(\ell)}(u,k)$ and $\mathbf{R}^{(\ell)}(u,k) \leftarrow 0$;

13   $\mathbf{Q}^{(L)} \leftarrow \mathbf{R}^{(L)}$;
14   $\hat{\mathbf{P}} \leftarrow \sum_{\ell=0}^{L} w_\ell \cdot \mathbf{D}^r \cdot \left( \mathbf{Q}^{(\ell)} + \sum_{t=0}^{\ell} \mathbf{S}^{(\ell-t)} \mathbf{R}^{(t)} \right)$;
15   **return** Embedding matrix $\hat{\mathbf{P}}_{n \times F}$ ;

---

### 3.1 The Bidirectional Propagation Algorithm

To reduce the time complexity, we propose approximating the Generalized PageRank matrix $\mathbf{P}$ with a localized bidirectional propagation algorithm from both the training/testing nodes and the feature vectors. Similar techniques have been used for computing probabilities in Markov Process [3]. Algorithm 1 illustrates the pseudo-code of the Bidirectional Propagation algorithm. The algorithm proceeds in three phases: Monte-Carlo Propagation (lines 1-5), Reverse Push Propagation (lines 6-13), and the combining phase (line 14).

**Monte Carlo Propagation from the training/testing nodes.** We start with a simple Monte-Carlo propagation (Lines 1-5 in Algorithm 1) from the training/testing nodes to estimate the transition probabilities. Given a graph $G$ and a training/testing node set $V_t$, we generate a number $n_r$ of random walks from each node $s \in V_t$, and record $\mathbf{S}^{(\ell)}(s, u)$, the fraction of random walks that visit node $u$ at the $\ell$-th steps. Note that $\mathbf{S}^{(\ell)} \in \mathcal{R}_{n \times n}$ is a sparse matrix with at most $|V_t| \cdot n_r$ non-zero entries. Since each random walk is independently sampled, we have that $\mathbf{S}^{(\ell)}$ is an unbiased estimator for the $\ell$-th transition probability matrix $\left(\mathbf{D}^{-1}\mathbf{A}\right)^{\ell}$. We also note that $\left(\mathbf{D}^{r-1}\mathbf{A}\mathbf{D}^{-r}\right)^{\ell} = \mathbf{D}^r \left(\mathbf{D}^{-1}\mathbf{A}\right)^{\ell} \mathbf{D}^{-r}$, which means we can use $\mathbf{D}^r\mathbf{S}^{(\ell)}\mathbf{D}^{-r}\mathbf{X}$ as an unbiased estimator for the $\ell$-th propagation matrix $\mathbf{T}^{(\ell)} = \left(\mathbf{D}^{r-1}\mathbf{A}\mathbf{D}^{-r}\right)^{\ell}\mathbf{X}$. Consequently, $\sum_{\ell=0}^{L} w_\ell \mathbf{D}^r\mathbf{S}^{(\ell)}\mathbf{D}^{-r}\mathbf{X}$ serves as an unbiased estimator for the Generalized PageRank Matrix $\mathbf{P}$. However, this estimation requires a large number of random walks from each training node and thus is infeasible for fully-supervised training on large graphs.

**Reverse Push Propagation from the feature vectors.** To reduce the variance of the Monte-Carlo estimator, we employ a deterministic Reverse Push Propagation (lines 6-13 in Algorithm 1) from the feature vectors. Given a feature matrix $\mathbf{X}$, the algorithm outputs two sparse matrices for each level $\ell = 0, \dots, L$: the reserve matrix $\mathbf{Q}^{(\ell)}$ that represents the probability mass to stay at level $\ell$, and the residue matrix $\mathbf{R}^{(\ell)}$ that represents the probability mass to be distributed beyond level $\ell$. We begin by setting the initial residue $\mathbf{R}^{(0)}$ as the degree normalized feature matrix $\mathbf{D}^{-r}\mathbf{X}$. We also perform column-normalization on $\mathbf{R}^{(0)}$ such that each dimension of $\mathbf{R}^{(0)}$ has the same $L_1$-norm. Starting from level $\ell = 0$, if the absolute value of the residue entry $\mathbf{R}^{(\ell)}(u, k)$ exceeds a threshold $r_{max}$, we increase the residue of each neighbor $v$ at level $\ell + 1$ by $\frac{\mathbf{R}^{(\ell)}(u,k)}{d(v)}$ and transfer the residue of $u$ to its reserve $\mathbf{Q}^{(\ell)}(u, k)$. Note that by maintaining a list of residue entries $\mathbf{R}^{(\ell)}(u, k)$ that exceed the threshold $r_{max}$, the above push operation can be done without going through every entry in $\mathbf{R}^{(\ell)}$. Finally, we transfer the non-zero residue $\mathbf{R}^{(L)}(u, k)$ of each node $u$ to its reserve at level $L$.

We will show that when Reverse Push Propagation terminates, the reserve matrix $\mathbf{Q}^{(\ell)}$ satisfies

$$\mathbf{T}^{(\ell)}(s, k) - d(s)^r \cdot (\ell + 1) \cdot r_{max} \leq \left(\mathbf{D}^r\mathbf{Q}^{(\ell)}\right)(s, k) \leq \mathbf{T}^{(\ell)}(s, k) \tag{3}$$

for each training/testing node $s \in V_t$ and feature dimension $k$. Recall that $\mathbf{T}^{(\ell)} = \left(\mathbf{D}^{r-1}\mathbf{A}\mathbf{D}^{-r}\right)^{\ell}\mathbf{X}$ is the $\ell$-th propagation matrix. This property implies that we can use $\sum_{\ell=0}^{L} w_\ell \mathbf{D}^r\mathbf{Q}^{(\ell)}$ to approximate the Generalized PageRank matrix $\mathbf{P} = \sum_{\ell=0}^{L} w_\ell \mathbf{T}^{(\ell)}$. However, there are two drawbacks to this approximation. First, this estimator is biased, which could potentially hurt the performance of the prediction. Secondly, the Reverse Push Propagation does not take advantage of the semi-supervised learning setting where the number of training nodes may be significantly less than the total number of nodes $n$.

**Combining Monte-Carlo and Reverse Push Propagation.** Finally, we combine the results from the Monte-Carlo and Reverse Push Propagation to form a more accurate unbiased estimator. In particular, we use the following equation to derive an approximation of the $\ell$-th propagation matrix $\mathbf{T}^{(\ell)} = \left(\mathbf{D}^{r-1}\mathbf{A}\mathbf{D}^{-r}\right)^{\ell}\mathbf{X}$:

$$\hat{\mathbf{T}}^{(\ell)} = \mathbf{D}^r \cdot \left(\mathbf{Q}^{(\ell)} + \sum_{t=0}^{\ell} \mathbf{S}^{(\ell-t)}\mathbf{R}^{(t)}\right). \tag{4}$$

As we shall prove in Section 3.2, $\hat{\mathbf{T}}^{(\ell)}$ is an unbiased estimator for $\mathbf{T}^{(\ell)} = \left(\mathbf{D}^{r-1}\mathbf{A}\mathbf{D}^{-r}\right)^{\ell}\mathbf{X}$. Consequently, we can use $\hat{\mathbf{P}} = \sum_{\ell=0}^{L} w_\ell \hat{\mathbf{T}}^{(\ell)} = \sum_{\ell=0}^{L} w_\ell \mathbf{D}^r \cdot \left(\mathbf{Q}^{(\ell)} + \sum_{t=0}^{\ell} \mathbf{S}^{(\ell-t)}\mathbf{R}^{(t)}\right)$ as an

unbiased estimator for the Generalized PageRank matrix $\mathbf{P}$ defined in equation (2). To see why equation (4) is a better approximation than the naive estimator $\mathbf{D}^r \mathbf{S}^{(\ell)} \mathbf{D}^{-r} \mathbf{X}$, note that each entry in the residue matrix $\mathbf{R}^{(t)}$ is bounded by a small real number $r_{max}$, which means the variance of the Monte-Carlo estimator is reduced by a factor of $r_{max}$. It is also worth mentioning that the two matrices $\mathbf{S}^{(\ell-t)}$ and $\mathbf{R}^{(t)}$ are sparse, so the time complexity of the matrix multiplication only depends on their numbers of non-zero entries.

### 3.2 Analysis

We now analyze the time complexity and the approximation quality of the Bidirectional Propagation algorithm. Due to the space limit, we defer all proofs in this section to the appendix. Recall that $|V_t|$ is the number of training/testing nodes, $n_r$ is the number of random walks per node , and $r_{max}$ is the push threshold. We assume $\mathbf{D}^{-r} \mathbf{X}$ is column-normalized, as described in Algorithm 1. We first present a Lemma that bounds the time complexity of the Bidirectional Propagation algorithm.

**Lemma 1** *The time complexity of Algorithm 1 is bounded by* $O\left(L|V_t|n_r F + \frac{LdF}{r_{max}}\right)$.

Intuitively, the $L|V_t|n_r F$ term represents the time for the Monte-Carlo propagation, and the $\frac{LdF}{r_{max}}$ term is the time for the Reverse Push propagation. Next, we will show how to set the number of random walks $n_r$ and the push threshold $r_{max}$ to obtain a satisfying approximation quality. In particular, the following technical Lemma states that the Reverse Push Propagation maintains an invariant during the push process.

**Lemma 2** *For any residue and reserve matrices* $\mathbf{Q}^{(\ell)}, \mathbf{R}^{(\ell)}, \ell = 0, \ldots, L$, *we have*

$$\mathbf{T}^{(\ell)} = \left(\mathbf{D}^{r-1}\mathbf{A}\mathbf{D}^{-r}\right)^{\ell} \mathbf{X} = \mathbf{D}^r \cdot \left(\mathbf{Q}^{(\ell)} + \sum_{t=0}^{\ell} \left(\mathbf{D}^{-1}\mathbf{A}\right)^{\ell-t} \mathbf{R}^{(t)}\right). \tag{5}$$

We note that the only difference between equations (4) and (5) is that we replace $\mathbf{D}^{-1}\mathbf{A}$ with $\mathbf{S}^{(\ell)}$ in equation (4). Recall that $\mathbf{S}^{(\ell)}$ is an unbiased estimator for the $\ell$-th transition probability matrix $\left(\mathbf{D}^{-1}\mathbf{A}\right)^{\ell}$. Therefore, Lemma 2 ensures that $\hat{\mathbf{T}}^{(\ell)}$ is also an unbiased estimator for $\mathbf{T}^{(\ell)}$. Consequently, $\hat{\mathbf{P}} = \sum_{\ell=0}^{L} w_\ell \hat{\mathbf{T}}^{(\ell)}$ is an unbiased estimator for the propagation matrix $\mathbf{P}$. Finally, to minimize the overall time complexity of Algorithm 1 in Lemma 1, the general principle is to balance the costs of the Monte-Carlo and the Reverse Push propagations. In particular, we have the following Theorem.

**Theorem 1** *By setting* $n_r = O\left(\frac{1}{\varepsilon}\sqrt{\frac{d \log n}{|V_t|}}\right)$ *and* $r_{max} = O\left(\varepsilon\sqrt{\frac{d}{|V_t|\log n}}\right)$, *Algorithm 1 produces an estimator* $\hat{\mathbf{P}}$ *of the propagation matrix* $\mathbf{P}$, *such that for any* $s \in V_t$ *and* $k = 0, \ldots, F - 1$, *the probability that* $\left|\hat{\mathbf{P}}(s,k) - \mathbf{P}(s,k)\right| \le d(s)^r \varepsilon$ *is at least* $1 - \frac{1}{n}$. *The time complexity is bounded by* $O\left(L|V_t|F + L\frac{\sqrt{|V_t|d \log n}}{\varepsilon}F\right)$.

For fully-supervised learning, we have $|V_t| = n$ and thus the time complexity of GBP becomes $O\left(LnF + L\frac{\sqrt{m \log n}}{\varepsilon}F\right)$. In practice, we can also make a trade-off between efficiency and accuracy by manipulating the push threshold $r_{max}$ and the number of walks $n_r$ .

**Parallelism of GBP.** The Bidirectional Propagation algorithm is embarrassingly parallelizable: We can generate the random walks on multiple nodes and perform Reverse Push on multiple features in parallel. After we obtain the Generalized PageRank matrix $\mathbf{P}$, it is trivially to construct mini-batches for training the neural networks.

## 4 Experiments

**Datasets.** We use seven open graph datasets with different size: three citation networks Cora, Citeser and Pubmed [25], a Protein-Protein interaction network PPI [11], a customer interaction

Table 2: Dataset statistics.

| Dataset | Task | Nodes | Edges | Features | Classes | Label rate |
|---|---|---|---|---|---|---|
| Cora | multi-class | 2,708 | 5,429 | 1,433 | 7 | 0.052 |
| Citeseer | multi-class | 3,327 | 4,732 | 3,703 | 6 | 0.036 |
| Pubmed | multi-class | 19,717 | 44,338 | 500 | 3 | 0.003 |
| PPI | multi-label | 56,944 | 818,716 | 50 | 121 | 0.79 |
| Yelp | multi-label | 716,847 | 6,977,410 | 300 | 100 | 0.75 |
| Amazon | multi-class | 2,449,029 | 61,859,140 | 100 | 47 | 0.70 |
| Friendster | multi-class | 65,608,366 | 1,806,067,135 | 100 (random) | 500 | 0.001 |

network Yelp [37], a co-purchasing networks Amazon [8] and a large social network Friendster [34]. Table 2 summarizes the statistics of the datasets. We first evaluate GBP's performance for transductive semi-supervised learning on the three popular citation networks (Cora, Citeseer, and Pubmed). Then we compare GBP with scalable GNN methods three medium to large graphs PPI, Yelp, Amazon in terms of inductive learning ability. Finally, we present the first empirical study of transductive semi-supervised on billion-scale network Friendster.

**Baselines and detailed setup.** We adopt three state-of-the-art GNN methods GCN [15], GAT [29], GDC [17] and APPNP [16] as the baselines for evaluation on small graphs. We also use one state-of-the-art scalable GNN from each of the three categories: LADIES (layer sampling) [40], GraphSAINT (graph sampling) [37], SGC and PPRGo (linear model) [30, 4].

We implement GBP in PyTorch and C++, and employ initial residual connection [12] across the hidden layers to facilitate training. For simplicity, we use the Personalized PageRank weights ($w_\ell = \alpha(1 - \alpha)^\ell$ for some hyperparameter $\alpha \in (0, 1)$). As we shall see, this weight sequence generally achieves satisfying results on graphs with real-world features. On the Friendster dataset, where the features are random noises, we use both Personalized PageRank and transition probability ($w_L = 1, w_0 =, \dots, = w_{L-1} = 0$) for GBP. We set $L = 4$ across all datasets. Table 3 summaries other hyper-parameters of GBP on different datasets. We use Adam optimizer to train our model, with a maximum of 1000 epochs and a learning rate of 0.01. For a fair comparison, we use the officially released code of each baseline (see the supplementary materials for URL and commit numbers) and perform a grid search to tune hyperparameters for models. All the experiments are conducted on a machine with an NVIDIA TITAN V GPU (12GB memory), Intel Xeon CPU (2.20 GHz), and 256GB of RAM.

Table 3: Hyper-parameters of GBP. $r_{max}$ is the Reverse Push Threshold, $w$ is the number of random walks from the training nodes, $w_\ell$ is the weight sequence, $r$ is the Laplacian parameter in the convolutional matrix $\mathbf{D}^{r-1}\mathbf{A}\mathbf{D}^{-r}$.

| Data | Dropout | Hidden dimension | $L_2$ | Batch size | $r_{max}$ | $w$ | $w_\ell$ | $r$ |
|---|---|---|---|---|---|---|---|---|
| Cora | 0.5 | 64 | 5e-4 | 16 | 1e-4 | 0 | $\alpha(1-\alpha)^\ell, \alpha = 0.1$ | 0.5 |
| Citeseer | 0.5 | 64 | 2e-1 | 64 | 1e-5 | 0 | $\alpha(1-\alpha)^\ell, \alpha = 0.15$ | 0.4 |
| Pubmed | 0.5 | 128 | 5e-4 | 16 | 1e-5 | 0 | $\alpha(1-\alpha)^\ell, \alpha = 0.05$ | 0.5 |
| PPI | 0.1 | 2048 | - | 2048 | 5e-7 | 0 | $\alpha(1-\alpha)^\ell, \alpha = 0.3$ | 0 |
| Yelp | 0.1 | 2048 | - | 30000 | 5e-7 | 0 | $\alpha(1-\alpha)^\ell, \alpha = 0.9$ | 0.3 |
| Amazon | 0.1 | 1024 | - | 100000 | 1e-7 | 0 | $\alpha(1-\alpha)^\ell, \alpha = 0.2$ | 0.2 |
| Friendster (PPR) | 0.1 | 128 | - | 2048 | 4e-8 | 10000 | $\alpha(1-\alpha)^\ell, \alpha = 0.1$ | 0.5 |
| Friendster | 0.1 | 128 | - | 2048 | 4e-8 | 10000 | $\{0, 0, 0, 1\}$ | 0.5 |

**Transductive learning on small graphs.** Table 4 shows the results for the semi-supervised transductive node classification task on the three small standard graphs Cora, Citeseer, and Pubmed. Following [15], we apply the standard fixed training/validation/testing split with 20 nodes per class for training, 500 nodes for validation and 1,000 nodes for testing. For each method, we set the number of hidden layers to 2 and take the mean accuracy with the standard deviation after ten runs. We observe that GBP outperforms APPNP (and consequently all other baselines) across all datasets. For the scalable GNNs, SGC is outperformed by the vanilla GCN due to the simplification [30]. On the other hand, the results of LADIES and GraphSAINT are also not at par with the non-scalable GNNs

Table 4: Results on Cora, Citeseer and Pubmed.

| Method | Cora | Citeseer | Pubmed |
|---|---|---|---|
| GCN | $81.5 \pm 0.6$ | $71.3 \pm 0.4$ | $79.1 \pm 0.4$ |
| GAT | $83.3 \pm 0.8$ | $71.9 \pm 0.7$ | $78.0 \pm 0.4$ |
| GDC | $83.3 \pm 0.2$ | $72.2 \pm 0.3$ | $78.6 \pm 0.4$ |
| APPNP | $83.3 \pm 0.3$ | $71.4 \pm 0.6$ | $80.1 \pm 0.2$ |
| SGC | $81.0 \pm 0.1$ | $71.8 \pm 0.1$ | $79.0 \pm 0.1$ |
| LADIES | $79.6 \pm 0.5$ | $68.6 \pm 0.3$ | $77.0 \pm 0.5$ |
| PPRGo | $82.4 \pm 0.2$ | $71.3 \pm 0.3$ | $80.0 \pm 0.4$ |
| GraphSAINT | $81.3 \pm 0.4$ | $70.5 \pm 0.4$ | $78.2 \pm 0.8$ |
| GBP | $\mathbf{83.9 \pm 0.7}$ | $\mathbf{72.9 \pm 0.5}$ | $\mathbf{80.6 \pm 0.4}$ |

such as GAT or APPNP, which suggests that the sampling technique alone might not be sufficient to achieve satisfying performance on small graphs.

Table 5: Results of inductive learning with scalable GNNs.

| | | 4-layer | | 6-layer | | 8-layer | |
|---|---|---|---|---|---|---|---|
| | | F1-score | Time (s) | F1-score | Time (s) | F1-score | Time (s) |
| PPI | SGC | $65.7 \pm 0.01$ | $\mathbf{76}$ | $62.4 \pm 0.01$ | 173 | $57.8 \pm 0.01$ | 295 |
| | LADIES | $57.9 \pm 0.30$ | 187 | $59.4 \pm 0.25$ | 206 | $57.4 \pm 0.24$ | 315 |
| | PPRGo | $61.5 \pm 0.13$ | 866 | $61.1 \pm 0.02$ | 1976 | $55.1 \pm 0.19$ | 1080 |
| | GraphSAINT | $99.2 \pm 0.05$ | 1291 | $\mathbf{99.4 \pm 0.03}$ | 1961 | $\mathbf{99.3 \pm 0.01}$ | 2615 |
| | GBP | $\mathbf{99.3 \pm 0.02}$ | 117 | $99.3 \pm 0.03$ | $\mathbf{167}$ | $99.3 \pm 0.01$ | $\mathbf{220}$ |
| Yelp | SGC | $41.5 \pm 0.21$ | 43 | $36.8 \pm 0.33$ | 70 | $34.8 \pm 0.52$ | 92 |
| | LADIES | $27.3 \pm 0.56$ | 34 | $28.5 \pm 0.97$ | 39 | $30.0 \pm 0.32$ | 51 |
| | PPRGo | $64.0 \pm 0.16$ | 550 | $63.7 \pm 0.71$ | 1215 | $63.4 \pm 0.49$ | 1665 |
| | GraphSAINT | $64.7 \pm 0.08$ | 712 | $62.0 \pm 0.10$ | 996 | $59.1 \pm 0.35$ | 1298 |
| | GBP | $\mathbf{65.4 \pm 0.03}$ | $\mathbf{19}$ | $\mathbf{65.5 \pm 0.03}$ | $\mathbf{30}$ | $\mathbf{65.4 \pm 0.05}$ | $\mathbf{37}$ |
| Amazon | SGC | $90.4 \pm 0.01$ | 233 | $89.9 \pm 0.03$ | 284 | $89.7 \pm 0.03$ | 342 |
| | LADIES | $85.4 \pm 0.14$ | 734 | $85.2 \pm 0.20$ | 784 | $84.6 \pm 0.09$ | 1421 |
| | PPRGo | $83.3 \pm 0.51$ | 2775 | $83.3 \pm 0.09$ | 5206 | $81.6 \pm 0.22$ | 9300 |
| | GraphSAINT | $\mathbf{91.5 \pm 0.01}$ | 957 | $91.3 \pm 0.05$ | 1228 | $91.4 \pm 0.05$ | 2618 |
| | GBP | $\mathbf{91.5 \pm 0.01}$ | $\mathbf{225}$ | $\mathbf{91.5 \pm 0.01}$ | $\mathbf{243}$ | $\mathbf{91.6 \pm 0.01}$ | $\mathbf{300}$ |

**Inductive learning on medium to large graphs.**    Table 5 reports the F1-score and running time (precomputation + training) of each method with various depths on three large datasets PPI, Yelp, and Amazon. For each dataset, we set the hidden dimension to be the same across all methods: 2048(PPI), 2048(Yelp), and 1024(Amazon). We use "fixed-partition" splits for each dataset, following [37, 8] (see the supplementary materials for further details). The critical observation is that GBP can achieve comparable performance as GraphSAINT does, with 5-10x less running time. This demonstrates the superiority of GBP's sub-linear time complexity. For PPRGo, it has a longer running time than other methods because of its expensive feature propagation per epoch. On the other hand, SGC and LADIES are also able to run faster than GraphSAINT; However, these two models' accuracy is not comparable to that of GraphSAINT and GBP.

Figure 1 shows the convergence rate of each method, in which the time for data loading, pre-processing, validation set evaluation, and model saving are excluded. We observe that the convergence rate of GBP and SGC is much faster than that of LADIES and GraphSAINT, which is a benefit from decoupling the feature propagation and the neural networks.

**Transductive semi-supervised learning on billion-scale graph Friendster.**    Finally, we perform the first empirical evaluation of scalable GNNs on a billion-scale graph Friendster. We extracted the top-500 ground-truth communities from [34] and use the community ids as the labels of each node. Note that one node may belong to multiple communities, in which case we pick the largest community as its label. The goal is to perform multi-class classification with only the graph structural information. This setting has been adapted in various works on community detection [19, 18, 35]. For each node, we generate a sparse random feature by randomly set one entry to be 1 in an $d$-dimensional all-zero vector. Note that even with a random feature matrix, GNNs are still able to extract structural

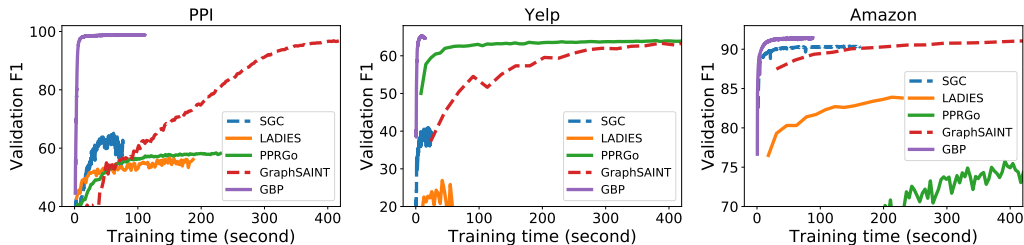

Figure 1: Convergence curves of 4-layer models.

information to perform the prediction [33]. Among the labeled nodes, we use 50,000 nodes (100 from each class) for training, 15,982 for validation, and 25,000 for testing. Table 6 report the running time and F1-score of each method with feature dimension $F = 10, 40, 70$ and 100. We omit GraphSAINT and LADIES as they run out of the 256 GB memory even with the dimension $d$ set to 10. We first observe that both GBP and SGC can capture the structural information with random features, while PPRGo and GBP(PPR) fail to converge. This is because Personalized PageRank emphasizes each node's original feature (with $w_0$ being the maximum weight among $w_0, \ldots, w_L$) and, yet, the original features of Friendster are random noises. We also point out that PPRGo starts to converge and achieves an F1-score of 0.15 in 4500 seconds when the feature dimension is increased to 10000. On the other hand, we observe that GBP can achieve a significantly higher F1-score with less running time. Notably, on this 500-class classification task, GBP is able to achieve an F1-score of 0.79 with less than half an hour.

Table 6: Results for semi-supervised learning on Friendster.

| Dimension | 10 | | 40 | | 70 | | 100 | |
|---|---|---|---|---|---|---|---|---|
| F1-score / Time | F1-score | Time | F1-score | Time | F1-score | Time | F1-score | Time |
| SGC | $2.0 \pm 0.27$ | 1130 | $12.9 \pm 0.01$ | 2930 | $27.1 \pm 0.01$ | 4549 | $40.2 \pm 0.01$ | 6379 |
| PPRGo | $1.6 \pm 0.01$ | - | $1.6 \pm 0.01$ | - | $1.6 \pm 0.01$ | - | $1.6 \pm 0.01$ | - |
| GBP(PPR) | $1.6 \pm 0.01$ | - | $1.6 \pm 0.01$ | - | $7.3 \pm 0.20$ | - | $7.3 \pm 0.12$ | - |
| **GBP** | $\mathbf{7.5 \pm 0.10}$ | **757** | $\mathbf{26.6 \pm 0.04}$ | **863** | $\mathbf{50.3 \pm 0.44}$ | **1392** | $\mathbf{79.7 \pm 0.32}$ | **1849** |

## 5  Conclusion

This paper presents GBP, a scalable GNN based on localized bidirectional propagation. Theoretically, GBP is the first method that achieves sub-linear time complexity for precomputation, training, and inference. The bidirectional propagation process computes a Generalized PageRank matrix that can express various existing graph convolutions. Extensive experiments on real-world graphs show that GBP obtains significant improvement over the state-of-the-art methods in terms of efficiency and performance. Furthermore, GBP is the first method that can scale to billion-edge networks on a single machine. For future work, an interesting direction is to extend GBP to heterogeneous networks.

## Broader Impact

The proposed GBP algorithm addresses the challenge of scaling GNNs on large graphs. We consider this algorithm a general technical and theoretical contribution, without any foreseeable specific impacts. For applications in bioinformatics, computer vision, and natural language processing, applying the GBP algorithm may improve the scalability of existing GNN models. We leave the exploration of other potential impacts to future work.

## Acknowledgments and Disclosure of Funding

Ji-Rong Wen was supported by National Natural Science Foundation of China (NSFC) No.61832017, and by Beijing Outstanding Young Scientist Program NO. BJJWZYJH012019100020098. Zhewei Wei was supported by NSFC No. 61972401 and No. 61932001, by the Fundamental Research Funds for the Central Universities and the Research Funds of Renmin University of China under Grant 18XNLG21, and by Alibaba Group through Alibaba Innovative Research Program. Ye Yuan was supported by NSFC No. 61932004 and No. 61622202, and by FRFCU No. N181605012. Xiaoyong Du was supported by NSFC No. U1711261.

## Footnotes

\*Zhewei Wei is the corresponding author. Work partially done at Gaoling School of Artificial Intelligence, Beijing Key Laboratory of Big Data Management and Analysis Methods, MOE Key Lab DEKE, Renmin University of China, and Pazhou Lab, Guangzhou, 510330, China.

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
