[Supplementary Material · supplementary.pdf]

# A  Proofs

We need the following Chernoff Bound for bounded i.i.d. random variables.

**Lemma 3 (Chernoff Bound [9])**  *Consider a set $\{x_i\}$ $(i \in [1, n_r])$ of i.i.d. random variables with mean $\mu$ and $x_i \in [0, r]$, we have*

$$\Pr\left[\left|\frac{1}{n_r}\sum_{i=1}^{n_r} x_i - \mu\right| \geq \varepsilon\right] \leq \exp\left(-\frac{n_r \cdot \varepsilon^2}{r\left(\frac{2}{3}\varepsilon + 2\mu\right)}\right). \tag{6}$$

## A.1  Proof of Lemma 1

Recall that in the Monte-Carlo Propagation phase of Algorithm 1, we first generate $n_r$ random walks of length $L$ for each training/testing node $s \in V_t$ to estimate the $\ell$-th transition probability matrix $\mathbf{S}^{(\ell)}$, $\ell = 0, \ldots, L$. Since the number of training/testing nodes is $|V_t|$, the total cost is bounded by $O(L|V_t|n_r)$. After deriving $\mathbf{S}^{(\ell)}$, we need to compute $\sum_{\ell=0}^{L} w_\ell \sum_{t=0}^{\ell} \mathbf{S}^{(\ell-t)}\mathbf{R}^{(t)}$ (line 14 in Algorithm 1). Since there are at most $O(|V_t| \cdot n_r)$ non-zero entries in each $\mathbf{S}^{(\ell)}$, the total cost can be bounded by $O(L|V_t|n_r F)$.

On the other hand, in the Reverse Push Propagation phase of Algorithm 1, we push the residue $\mathbf{R}^{(\ell)}(u, k)$ of node $u$ to its neighbors whenever $\left|\mathbf{R}^{(\ell)}(u, k)\right| > r_{max}$, $k = 0, \ldots, F - 1$. For random features, the average cost for this push operation is $d$, the average degree of the graph. We also observe that for a given level $\ell$ and a given feature dimension $k$, there are at most $1/r_{max}$ nodes with residues larger than $r_{max}$. Consequently, the cost of Reverse Push for a given level $\ell$ and a given feature dimension $k$ is $\frac{d}{r_{max}}$. Summing up $\ell = 0, \ldots, L - 1$ and $k = 0, \ldots, F - 1$, and the Lemma follows.

## A.2  Proof of Lemma 2

Let $\mathcal{RHS}$ denote the right hand side of equation (5); We prove the Lemma by induction. Recall that in Algorithm 1, we initialize $\mathbf{Q}^{(t)} = 0$ and $\mathbf{R}^{(t)} = 0$ for $t = 0, \ldots, \ell$, and $\mathbf{R}^{(0)} = \mathbf{D}^{-r}\mathbf{X}$. Consequently, we have

$$\mathcal{RHS} = \mathbf{D}^r \left(\mathbf{D}^{-1}\mathbf{A}\right)^\ell \mathbf{R}^{(0)} = \mathbf{D}^r \left(\mathbf{D}^{-1}\mathbf{A}\right)^\ell \mathbf{D}^{-r}\mathbf{X} = \left(\mathbf{D}^{r-1}\mathbf{A}\mathbf{D}^{-r}\right)^\ell \mathbf{X} = \mathbf{T}^{(\ell)},$$

which is true by definition. Assuming Equation (5) holds at some stage, we will show that the invariant still holds after a push operation on node $u$. More specifically, let $\mathbf{I}_{uk} \in \mathcal{R}^{n \times F}$ denote the matrix with entry at $(u, k)$ setting to 1 and the rest setting to zero. Consider a push operation on $u \in V$ and $k \in 0, \ldots, F - 1$ with $|\mathbf{R}^{(t)}(u, k)| > r_{max}$. We have two cases:

(1) If $t \leq \ell - 1$, we have $\mathbf{R}^{(t)}$ is decremented by $\mathbf{R}^{(t)}(u, k) \cdot \mathbf{I}_{uk}$ and $\mathbf{R}^{(t+1)}$ is incremented by $\frac{\mathbf{R}^{(t)}(u,k)}{d(v)} \cdot \mathbf{I}_{vk}$ for each $v \in N(u)$. Consequently, we have

$$\mathcal{RHS} = \mathbf{T}^{(\ell)} + \mathbf{D}^r \cdot \left(\mathbf{D}^{-1}\mathbf{A}\right)^{\ell-t} \left(-\mathbf{R}^{(t)}(u,k) \cdot \mathbf{I}_{uk}\right) + \mathbf{D}^r(\mathbf{D}^{-1}\mathbf{A})^{\ell-t-1} \cdot \sum_{v \in \mathcal{N}(u)} \frac{\mathbf{R}^{(t)}(u,k)}{d(v)} \cdot \mathbf{I}_{vk}$$

$$= \mathbf{T}^{(\ell)} + \mathbf{R}^{(t)}(u,k) \cdot \mathbf{D}^r(\mathbf{D}^{-1}\mathbf{A})^{\ell-t-1} \cdot \left(\sum_{v \in \mathcal{N}(u)} \frac{1}{d(v)} \cdot \mathbf{I}_{vk} - \mathbf{D}^{-1}\mathbf{A}\mathbf{I}_{uk}\right)$$

$$= \mathbf{T}^{(\ell)} + \mathbf{R}^{(t)}(u,k) \cdot \mathbf{D}^r(\mathbf{D}^{-1}\mathbf{A})^{\ell-t-1}\mathbf{0} = \mathbf{T}^{(\ell)}.$$

For the second last equation, we use the fact that $\sum_{v \in \mathcal{N}(u)} \frac{1}{d(v)} \cdot \mathbf{I}_{vk} = \mathbf{D}^{-1}\mathbf{A}\mathbf{I}_{uk}$.

(2) If $t = \ell$, we have $\mathbf{R}^{(\ell)}$ is decremented by $\mathbf{R}^{(\ell)}(u, k) \cdot \mathbf{I}_{uk}$ and $\mathbf{Q}^{(\ell)}$ is incremented by $\mathbf{R}^{(\ell)}(u, k) \cdot \mathbf{I}_{uk}$. Consequently, we have

$$\mathcal{RHS} = \mathbf{T}^{(\ell)} + \mathbf{D}^r \cdot \left(-\mathbf{R}^{(\ell)}(u,k) \cdot \mathbf{I}_{uk}\right) + \mathbf{D}^r \cdot \left(\mathbf{R}^{(\ell)}(u,k) \cdot \mathbf{I}_{uk}\right) = \mathbf{T}^{(\ell)} + \mathbf{D}^r \cdot \mathbf{0} = \mathbf{T}^{(\ell)}.$$

Therefore, the induciton holds, and the Lemma follows.

## A.3 Proof of Theorem 1

To show that Algorithm 1 achieves the desired accuracy, recall that equation (4) is an unbiased estimator for the $\ell$-th propagation matrix $\mathbf{T}^{(\ell)}$. We also observe each entry in residue matrix $\mathbf{R}^{(\ell)}$ derived by the reserve push propagation is bounded by $r_{max}$, and we multiply $\mathbf{D}^r$ to the estimator $\mathbf{Q}^{(\ell)} + \sum_{t=0}^{\ell-1} \mathbf{S}^{(\ell-t)} \mathbf{R}^{(t)}$, it follows the random variable of each random walk from node $s \in V_t$ is bounded by $d(s)^r \cdot r_{max}$. By Chernoff Bound (Lemma 3), we have

$$\Pr\left[\left|\hat{\mathbf{T}}^{(\ell)}(s,k) - \mathbf{T}^{(\ell)}(s,k)\right| \geq d(s)^r \varepsilon\right] \leq \exp\left(-\frac{n_r \cdot d(s)^r \cdot \varepsilon^2}{r_{max}\left(\frac{2}{3}\varepsilon + 2\mu\right)}\right) \leq \exp\left(-\frac{n_r \cdot \varepsilon^2}{r_{max}\left(\frac{2}{3}\varepsilon + 2\mu\right)}\right).$$

Where $\mu = \mathbf{T}^{(\ell)}(s,k)$. By setting $n_r = O\left(\frac{r_{max}\log n}{\varepsilon^2}\right)$, we have

$$\Pr\left[\left|\hat{\mathbf{T}}^{(\ell)}(s,k) - \mathbf{T}^{(\ell)}(s,k)\right| \geq d(s)^r \varepsilon\right] \leq \exp\left(-\frac{\log n}{\frac{2}{3}\varepsilon + 2\mu}\right) = O\left(\frac{1}{n}\right).$$

By Lemma 1, the time complexity of the Monte-Carlo Propagation is $O(L|V_t|n_r F)$, and the time complexity of the Reserve Push Propagation is $O(L\frac{d}{r_{max}}F)$. By setting $n_r = O(\frac{r_{max}\log n}{\varepsilon^2})$, the time complexity of Algorithm 1 can be express as

$$O\left(L|V_t|F + L|V_t|\frac{r_{max}\log n}{\varepsilon^2}F + L\frac{d}{r_{max}}F\right).$$

We observe that the above complexity is minimized when $L|V_t|\frac{r_{max}\log n}{\varepsilon^2}F = L\frac{d}{r_{max}}F$, which implies that

$$r_{max} = \sqrt{\varepsilon^2 \frac{d}{|V_t|\log n}} = \varepsilon\sqrt{\frac{d}{|V_t|\log n}}.$$

Therefore, the number of random walks per node $n_r$ can be expressed as

$$n_r = \frac{\log n}{\varepsilon^2} \cdot \varepsilon\sqrt{\frac{d}{|V_t|\log n}} = \frac{1}{\varepsilon}\sqrt{\frac{d\log n}{|V_t|}}.$$

Finally, the total time complexity of Algorithm 1 is bounded

$$O\left(L|V_t|F + L|V_t|\frac{r_{max}\log n}{\varepsilon^2}F + L\frac{d}{r_{max}}F\right) = O\left(L|V_t|F + L\frac{\sqrt{|V_t|d\log n}}{\varepsilon}F\right),$$

and the Theorem follows.

# B   Additional experimental results

## B.1   Comparison of inference time

Figure 2 shows the inference time of each method. We observe that in terms of the inference time, the three linear models, SGC, PPRGo and GBP, have a significant advantage over the two sampling-based models, LADIES and GraphSAINT.

## B.2   Additional details in experimental setup

Table 7 summarizes URLs and commit numbers of baseline codes.

Figure 2: Inference time of 6-layers models on the entire test graph.

Table 7: URLs of baseline codes.

| Methods | URL | Commit |
|---------|-----|--------|
| GCN | `https://github.com/rusty1s/pytorch_geometric` | 5692a8 |
| GAT | `https://github.com/rusty1s/pytorch_geometric` | 5692a8 |
| APPNP | `https://github.com/rusty1s/pytorch_geometric` | 5692a8 |
| GDC | `https://github.com/klicperajo/gdc` | 14333f |
| SGC | `https://github.com/Tiiiger/SGC` | 6c450f |
| LADIES | `https://github.com/acbull/LADIES` | c7f987 |
| PPRGo | `https://github.com/TUM-DAML/pprgo_pytorch` | d9f991 |
| GraphSAINT | `https://github.com/GraphSAINT/GraphSAINT` | cd31c3 |