[Reviews · NeurIPS 2020]

Review 1

Summary and Contributions: This paper draws inspiration from previous GNN work, which separates prediction from node features from the actual message passing process. It uses the Generalized PageRank matrix to express this decoupling and then proposes to use Monte Carlo methods to approximate the propagation dynamics. It then introduces the Reverse Push Propagation algorithm, a deterministic algorithm to reduce the variance of the MC estimates. They then show results on a number of standard baselines, as well as convergence curves indicating much faster convergence over the PPI, Yelp, and Amazon datasets, as well as an evaluation on a real-world graph with millions of nodes. ** Update ** I've read the other reviewers' feedback and author response and decide to keep my score the same.

Strengths: - The paper tackles an important problem of scaling up GNNs to train efficiently on larger datasets. The method seems simple and straightforward to implement, and achieves competitive results on a number of graph baselines. They also show convergence curves showing that their method can train much faster than the baselines and achieve competitive performance. - Their model is simple to implement and extremely parallelizable. - They show that their method can impressively scale on a real-life graph with millions of nodes.

Weaknesses: - I'm not 100% convinced of the novelty of the work. It seems like most of the ideas here (generalized PageRank, separation of prediction and propagation, Monte Carlo approximation of transition probabilities) are drawn from previous work, with the main contribution the Reverse Push Propagation section. - I think the experimental section is a bit thin. It would be interesting to see an analysis of the performance curves as you tradeoff r_max and the number of walks. I'd also be curious for an explanation of why the value of r_max is chosen to be smaller for the larger datasets. I'm also curious how the model performs without any Monte Carlo approximations. Right now the results section doesn't contain much analysis, just a presentation of the results on 3 different tiers of datasets. - Some experimental details are bit lacking; for instance, I couldn't find any info about how you chose the values for r (the supplementary material lists the values used but not how they were chosen) or how you choose the weights w_l in equation (2). Because of this concern, I checked 'not reproducible' below.

Correctness: The empirical methodology might benefit from adopting some of the evaluation methods here: https://arxiv.org/pdf/1811.05868.pdf in order to be more convincing.

Clarity: The paper is mostly clearly written. The section on Reverse Push Propagation starting at line 165 could use a bit more introduction and a more intuitive explanation of how the algorithm works / the motivation before diving into the details.

Relation to Prior Work: The paper does a good job of discussing previous work and describing how their methods build on these works, and they provide a detailed analysis of the runtime / drawbacks of previous work when applied to larger graphs.

Reproducibility: No

Additional Feedback: - I believe the end of the equation in line 190 is missing a w_l term.


Review 2

Summary and Contributions: The paper introduces a Graph Neural Network via Bidirectional Propagation (GBP) that achieves sub-linear time complexity and is therefore suited for operating on large-scale graphs. The GBP pre-computation phase is based on a Monte Carlo propagation scheme, followed by a reverse push propagation. This information is used to obtain an unbiased estimator for the Generalized PageRank matrix, which can be used in conjunction with an MLP to obtain node predictions for mini-batches. The authors compare their approach to the related work by analyzing their theoretical time complexity, describe their method in detail, analyze its approximation quality, and perform an extensive emperical study on 7 different datasets on which they show impressive and fast performance. One of the datasets is newly introduced and contains around 65 million nodes and 1.8 billion edges. The authors show that they can train the proposed model on this billion-scale graph dataset in around 100 minutes.

Strengths: The paper is mostly well-written and introduces an interesting approach for scaling up Graph Neural Networks with potentially high impact in the graph community. Scalability of GNNs is one of their major issues, and so I see this paper as a welcome contribution. All claims made in the paper are supported by an in-depth time complexity and approximation quality analysis.

Weaknesses: In my honest opinion, the proprosed approach has two limitations: 1. GBP relies on decoupling propgations and predictions, which may result in less expressive GNNs. However, this seems to only be a small limitation since the authors show that the proposed approach is rather general and performance is very competitive compared to the related work. 2. GBP cannot seem to handle edge features or edge weights.

Correctness: yes

Clarity: yes

Relation to Prior Work: yes

Reproducibility: Yes

Additional Feedback: Major comments: ============= * Authors should discuss the limitations of their approach, e.g.: * Can one say how much performance is lost (if any) due to decoupling predictions from propgations, e.g. via ablation studies? * The incorporation of edge features seems to be non-trivial. Is GBP potentially suited for incorporating those? * Table 1 should, in addition to reporting training and inference time complexities, discuss the memory complexity of GBP in comparison to the related work. * SGC results seem not to be very well fine-tuned. For example, SGC can achieve competitive performance on PPI when increasing the feature dimensionality to 1024 or 2048. Did the authors tune the reported baselines? Minor comments/questions: ===================== * Line 40: "In theory, this complexity is undesirable for scalable GNNs": The authors should discuss this in more depth. In the end, an algorithm that is linear in the number of edges seems pretty scalable to me. * Line 72: "... feature transformation can be performed in significantly less time due to better parallelism of dense-dense matrix multiplications": I find this statement slightly misleading. While I agree that sparse-dense matrix multiplication is usually not as fast as dense-dense matrix multiplication, I feel like this is not the limiting factor for achieving scalability. IMO, scalability limitations arise from neighborhood explosion and memory consumption of storing activations. Authors should discuss this problem in more detail. * Line 98: "In the worst case, the number of clusters in the graph is 1, and Cluster-GCN essentially becomes vanilla GCN in terms of time complexity": I feel like this is a rather unfair statement regarding Cluster-GCN, since the number of partitions is a hyperparameter that should be incorporated when comparing time complexities. * Line 101: "Given a sampled node u, the probability that a neighbor of u is also sampled is b/n": Note that this is only true for the GraphSAINT node sampling technique, but not for the proposed edge or random walk sampling techniques. * Table 1: VRGCN is missing in Table 1, which has, e.g., a very fast inference time with the disadvantage of a high memory footprint. * Line 116: "GIN ... shows that GNNs are not better than the Weisfeiler-Lehman test in distinguishing graph structures." Since the authors mentioned that, how does GBP relate to that? * Line 204: "the L|V_t|wF term represents the time for the Monte-Carlo propagation": Why is the time complexity of Monte-Carlo propagation related to the feature dimensionality F? * Line 281: "We omit GraphSAINT and LADIES as they run out of the 256 GB memory even with the dimension set to 10": While this is not the fault of the authors, I am wondering why exactly this is the case. I do not see any limitations in both methods that should prevent them to scale to millions of nodes (even when training time might be slower). Can the authors elaborate on that? ============ Post rebuttal comments ================= I first want to thank the authors for their rebuttal, which addresses most concerns properly. However, my initial rating has been a bit too high given that the presented work provides an extension to the KDD'20 work "Scaling Graph Neural Networks with approximate PageRank" on how to achieve sub-linear time complexity. I feel that the authors should relate the presented approach more to this work, and actually also compare against it in the experimental evaluation. Therefore, I have decreased my score by one point.


Review 3

Summary and Contributions: In this paper, the authors propose to use generalized page rank from [19] for the feature precomputation of GNN. It shows the method has consistently perform better than available baselines and it demonstrates the first experiment of GNN (its variant) on billion-scale graph.

Strengths: 1. Using generalized pagerank is novel and timely for GNN precomputation 2. The estimation methods are scalable 3. Results show the promising of the proposed approach in terms of both accuracy and scalability

Weaknesses: 1. the weight coefficient w_l in Equation (2) is not discussed in the experiment study. I cannot find how it is defined. Note that the parameters can be important for the performance study 2. KDD 2020 has a new work: Scaling Graph Neural Networks with approxmiate pagerank. It is published after the submission. But the authors may need to cite the work, and clarify the difference. 3. The overall F1 improvement over APPNP is quite small (may not be significant)

Correctness: The experimental study didn't discuss how the weight coefficient w_l is defined. The authors do provide the source code which include the parameters. But none-the-less, I think those are important parameters for the evaluation, which should be discussed

Clarity: Yes. It is rather easy to read

Relation to Prior Work: yes. The authors provide a nice discussion on the prior works

Reproducibility: Yes

Additional Feedback:


Review 4

Summary and Contributions: This paper proposes a graph neural network method by using a localized bidirectional propagation process to improve the scalability of graph neural network. The authors also give the theoretical analysis of the proposed method and previous methods. Experiments are designed to show the effectiveness of the proposed method.

Strengths: (1). The paper is well written and easy to follow. (2). The authors demonstrate the effectiveness and theoretical analysis of the proposed method. Furthermore, the experiments show that the proposed method can achieves state-of-the-art performance.

Weaknesses: (1). The proposed BPA can be treated as a solution of a special case of GDC. Hence the GDC is a very important baseline. Although the time complexity of GDC[1] is O(n^2), GDC can use the sparsity of the GDC. Hence, in real applications, GDC can also achieve promising performance. The authors should compare the proposed method with GDC at least. [1]. Klicpera J., Weissenberger S. et al. Diffusion improves graph learning. NIPS, 2019. (2). The authors should finetune the baselines carefully. For example, the sampling numbers of each layer for LADIES are variational and the experimental performance can be improved for these baselines. But in the paper, I cannot find whether the authors finetune these parameters.

Correctness: yes.

Clarity: yes.

Relation to Prior Work: yes

Reproducibility: Yes

Additional Feedback: please refer to weakness. Especially for comparison with GDC.

[Author Response · NeurIPS 2020]

**R1 W1**: The main contribution is the combination of the Monte-Carlo Propagation and the Reverse Push Propagation,
which is the key to achieve sub-linear complexity. Furthermore, the proposed GBP algorithm is the first method that
achieves good scalability and performance on the billion-scale dataset Friendster.

**R1 W2**: The value of $r_{max}$ actually depends on the sparsity of the feature matrix $\mathbf{X}$. The features of Cora, Citeseer,
and Pubmed are (sparse) bag-of-words features and thus can tolerate large error. The features of other datasets are dense
representations, and thus we have to set $r_{max}$ to be small for an acceptable error.

According to Theorem 1, the number of random walks per node $w$ is automatically determined by $r_{max}$. In practice, for
full-supervised learning tasks (PPI, Yelp, and Amazon) where most nodes are training nodes, we simply set $w = 0$ and
only rely on the results of the Reverse Push Propagation. For semi-supervised tasks on Friendster, we set $w = 10,000$,
which significantly reduces the Reverse Push Propagation error and improves the model accuracy.

**R1 W3**: The default value of $r$ is $1/2$, and we perform a grid search to find the best value for $r$. In most cases, setting
$r = 1/2$ can achieve satisfying results. Line 436 presents the choice of $w_\ell$, please see **R3 W1** for a detailed discussion.

**R1 Correctness**: We will use the suggested evaluation methods in the final version of the paper.

**R2 W1 & Major comment 1**: Decoupling prediction and propagation sometimes improve performance (see the
ablation study on small datasets in the APPNP paper). We will include an ablation study on the larger datasets.

**R2 W2 & Major comment 2**: GBP can support edges weights by treating the adjacency matrix $\mathbf{A}$ as a weighted
matrix. Supporting edge features, however, is beyond the scope of this paper.

**R2 Major comment 3**: We follow the original setting of SGC to compute the prediction $\mathbf{Z} = SoftMax(\mathbf{S}^K\mathbf{X}\mathbf{W})$,
which means there is no hidden layer. We will include the results of SGC running with a multi-layer network.

**R2 Minor comments 1-8**: Thanks for these insightful comments! Our response: 1) For GNN, the $O(m)$ term is
multiplied by $L$ and $F$, so it makes sense to optimize the $O(m)$ term. 2) We agree that the scalability limitations
arise from neighborhood explosion and memory consumption of storing activations. Line 72 is merely a theoretical
argument to show why we are not trying to improve the $O(LnF^2)$ term. 3) We will include the number of clusters as a
hyperparameter for Cluster-GCN and analyze its complexity in the Stochastic Block Model. 4) We only analyzed the
complexity of GraphSAINT with node sampling for simplicity. 5) We will include VRGCN in Table 1. 6) To the best of
our knowledge, there is no study on the expressiveness of the PageRank-based GNNs in terms of the Weisfeiler-Lehman
test. 7) We combine the results of the Monte-Carlo Propagation with each Reverse Propagation vectors, leading to the
extra $F$ term. 8) The technical reason is that the codes of LADIES and GraphSAINT store multiple views of the graphs
to enable efficient subgraph sampling, which leads to memory overflow when the graph size is large.

**R3 W1 & R1 W3**: At line 436, we mention that GBP uses Personalized PageRank (PPR, $w_\ell = \alpha(1 - \alpha)^\ell$) for all
datasets other than Friendster. In fact, we can simply use PPR for datasets with real-world features. Friendster is a rare
exception, where we have to set the $w_\ell$ to be the $L$-th transition probability ($w_L = 1, w_0 =, \ldots, = w_{L-1} = 0$). This is
because PPR emphasizes each node's original feature (with $w_0$ being the maximum weight among $w_0, w_1, \ldots$) and, yet,
the original feature of Friendster is random noise. We will include the above discussion in the final version of the paper.

**R3 W2**: We will include a discussion on PPRGo (KDD2020). Note that PPRGo precomputes the approximate
Personalized PageRank (PPR) matrix $\mathbf{S}$ and uses matrix multiplication to calculate the propagation matrix $\mathbf{S} \cdot \mathbf{X}$. In
contrast, GBP estimates $\mathbf{S} \cdot \mathbf{X}$ by the reverse push from $\mathbf{X}$ and thus avoids the computation of $\mathbf{S}$.

**R3 W3**: The main focus of this paper is to improve the scalability of GNN. APPNP is unable to scale on large graphs.

**R4 W1**: As suggested, we present the performance of GDC on the three small datasets in Table 1, which is similar
to that of GBP. However, GDC is unable to run on the larger datasets due to its $O(n^2)$ space/time complexity. Note
that GDC performs the sparsification operation after computing the $n \times n$ diffusion matrix, which inherently leads to
the $O(n^2)$ space/time complexity. In fact, one of the main contributions of this paper is to improve the scalability of
PageRank-based GNN such as APPNP and GDC.

Table 1: GDC on three small datasets.

| Method | Cora | Citeseer | Pubmed |
|---|---|---|---|
| GDC | $83.3 \pm 0.2$ | $72.2 \pm 0.3$ | $78.6 \pm 0.4$ |
| GBP | $\mathbf{83.9 \pm 0.7}$ | $\mathbf{72.9 \pm 0.5}$ | $\mathbf{80.6 \pm 0.4}$ |

Table 2: LADIES with varying number of samples per node.

| #Samples | 256 | 512 | 1024 | 2048 |
|---|---|---|---|---|
| PPI | 57.8 (205) | **59.4** (206) | 57.5 (215) | 58.1 (206) |
| Yelp | 27.1 (34) | 28.5 (39) | 26.1 (44) | **29.6** (37) |
| Amazon | 84.8 (793) | **85.2** (784) | 85.1 (787) | 84.8 (799) |

**R4 W2**: We present the performance of LADIES with a varying number of samples in Table 2. There is no significant
change in performance and training time as we tune the number of samples per node from 256 to 2048.

[Meta-Review · NeurIPS 2020]

This paper presents a solution to learn GNNs on large scale graphs efficiently, gaining substantial speed improvement without sacrificing model quality. A couple reviewers raised the issue that there is a parallel work published at KDD using a similar approach solving the same problem, however since the KDD paper is published after the submission of this paper this shouldn’t be used as evidence to discredit the contribution of this paper. On the other side I do hope the authors can include a discussion of this related work and improve the clarity issues raised by the reviewers in the final version.